# Traversing a Tightrope between Ecumenism and Exclusivism: The Intertwined History of South Africa's Dutch Reformed Church and the Church of Central Africa Presbyterian in Nyasaland (Malawi)

Retief Müller [†]

Nagel Institute for the Study of World Christianity, Calvin University, Grand Rapids, MI 49546, USA; rm38@calvin.edu

[†] Retief Müller is also research associate at Stellenbosch University's Faculty of Theology in the discipline group of systematic theology and ecclesiology.

**Abstract:** During the first few decades of the 20th century, the Nkhoma mission of the Dutch Reformed Church of South Africa became involved in an ecumenical venture that was initiated by the Church of Scotland's Blantyre mission, and the Free Church of Scotland's Livingstonia mission in central Africa. Geographically sandwiched between these two Scots missions in Nyasaland (presently Malawi) was Nkhoma in the central region of the country. During a period of history when the DRC in South Africa had begun to regressively disengage from ecumenical entanglements in order to focus on its developing discourse of Afrikaner Christian nationalism, this venture in ecumenism by one of its foreign missions was a remarkable anomaly. Yet, as this article illustrates, the ecumenical project as finalized at a conference in 1924 was characterized by controversy and nearly became derailed as a result of the intransigence of white DRC missionaries on the subject of eating together with black colleagues at a communal table. Negotiations proceeded and somehow ended in church unity despite the DRC's missionaries' objection to communal eating. After the merger of the synods of Blantyre, Nkhoma and Livingstonia into the unified CCAP, distinct regional differences remained, long after the colonial missionaries departed. In terms of its theological predisposition, especially on the hierarchy of social relations, the Nkhoma synod remains much more conservative than both of its neighboring synods in the CCAP to the south and north. Race is no longer a matter of division. More recently, it has been gender, and especially the issue of women's ordination to ministry, which has been affirmed by both Blantyre and Livingstonia, but resisted by the Nkhoma synod. Back in South Africa, these events similarly had an impact on church history and theological debate, but in a completely different direction. As the theology of Afrikaner Christian nationalism and eventually apartheid came into positions of power in the 1940s, the DRC's Nkhoma mission in Malawi found itself in a position of vulnerability and suspicion. The very fact of its participation in an ecumenical project involving 'liberal' Scots in the formation of an indigenous black church was an intolerable digression from the normative separatism that was the hallmark of the DRC under apartheid. Hence, this article focuses on the variegated entanglements of Reformed Church history, mission history, theology and politics in two different 20th-century African contexts, Malawi and South Africa.

**Keywords:** Afrikaner; apartheid; mission; Malawi; nationalism; South Africa

## 1. Introduction

When the German theologian, Martin Kähler made his frequently quoted claim describing mission as 'the mother of theology' (Kähler 1971, p. 190), the point was to bring about a reorientation regarding understanding the church as an essentially missionary institution in its earliest foundation. In many ways, the mid-20th-century German, and generally Western, church to which Kähler spoke remained stuck in a Christendom paradigm,

where an unstated general assumption perhaps considered theology not so much as a 'supporting manifestation of the Christian mission' but as a luxury discourse of a self-confident religious enterprise (Kähler 1971, p. 189). Albeit for a somewhat different purpose than what Kähler had in mind, this article takes his assertion regarding the close, perhaps even symbiotic, relationship between mission and theology as a theoretical point of departure. With reference to a specific case study involving the history of Christianity in southern and central Africa, this article will seek to tease out some of the complexities that were involved in this relationship in the colonial and early apartheid era.

The complex interplay between mission, church history, theology and politics is well illustrated by the case of the Dutch Reformed Church (DRC) of South Africa and its missionary and ecumenical interactions in Nyasaland/Malawi. What makes this case particularly noteworthy is its role within the development of the ideology of apartheid in the early to mid-20th century. (This article is broadly based on themes emerging from my forthcoming book (Müller), *The Scots Afrikaners: identity politics and intertwined religious cultures in southern and central Africa* (Edinburgh University Press)) The focus in this article is particularly on the role of one specific Scottish immigrant family in these parts, who together with their influence sphere exercised an outsized role in terms of the mission and policies of the DRC.

A general thesis argued here is that mission coupled with ecumenism might serve to counteract harmful nationalist discourses, but when mission finds itself coupled with nationalism instead, then a theology of separateness or exclusivism is virtually the inevitable outcome. Focusing on this particular case study, this article would thus substantiate this special edition's central claim, which challenges the division between the history of Christianity, the history of Christian mission, and the history of theology. The special edition purports to argue instead for a constant conversation between these fields and in support of that, this article will give a historical example proceeding from an approach that takes the conversation between these fields as basis of enquiry.

In order to illustrate this particular case to full effect, it would be necessary to go as far back as the early 19th century. By 1806, Britain had definitively taken over colonial rule at the Cape colony when Dutch reign ceased with the demise of the Batavian Republic in Europe (Boas and Weiskopf 1973). For the preceding, nearly, century and a half, the Cape was loosely controlled by the Dutch East India Company. The Dutch Reformed Church was the established church in the colony. With the advent of British rule in the early 19th century, the new colonial governor, Lord Charles Somerset, initiated a policy of anglicization among the populace (MacKenzie and Dalziel 2012, p. 267). One strategy was to populate and thereby anglicize vacancies within the Dutch church with, perhaps ironically, Scottish ministers. Theologically, the Scots were acceptable to the populace due to the fact that as Presbyterians, they came from the same broad Calvinist tradition to which the Dutch Reformed Church also belonged (Sass 1956, pp. 14–15). Culturally, Scots had a reputation for successfully assimilating to other cultures and ethnicities throughout the British Empire. This was no exception in South Africa (MacKenzie and Dalziel 2012, p. 55). A consequence of this policy, aided by the many pastoral vacancies in an expanding colonial frontier, was that Scots soon constituted a powerful factor in the DRC, even outsized in terms of the influence they ended up wielding.

## 2. Scots Missionaries in a Dutch Church

However noteworthy the 19th-century Scottish influence was, which was even termed a salvation of sorts by historian John Mackenzie (MacKenzie and Dalziel 2012, p. 9), the reason for mentioning this development relates to the perceived motivations of the Scots who availed themselves for serving in the Dutch Reformed Church. Before addressing that question directly, let me emphasize two significant historical factors playing roles in this context.

The first concerns the fact that within colonial society in 19th-century South Africa, there were strong class divisions showing up primarily along cultural linguistic lines. White

'English' colonists whose livelihoods often revolved around mercantile, governmental or missionary interests tended to belong to rather different educational and societal strata than the rural Dutch farmers. The former tended to view the latter in a stereotypically negative light (Johns 2013). The Dutch were generally poorly educated landholders with a vested interest in the maintenance of slavery as an institution, whilst the former, exemplified by someone like Dr John Philip, the local superintendent of the London Missionary Society, tended much more strongly towards abolitionist ideals (MacKenzie and Dalziel 2012, pp. 76–77). This is not to suggest that all of the British colonists were abolitionists or much interested in the lot of the indigenous population, generally. To the contrary, as was typical of colonial societies overall, the indigenous Africans, in this case, remained distinctly other from the colonial point of view. Racism rather than equalization was the order of the day. There would likely be nothing surprising about this assertion to any informed reader. Colonial societies thrived on inequality virtually by definition. However, one aspect that complexified the South African case is the fact that within the colonist population there existed from early on these rather stark divisions between the 'English' and 'Dutch' sectors. In fact, in important and seditious ways, the Dutch were evaluated as on a comparable level as the indigenous Africans from the point of view of the British who maintained the political and economic power in most of Southern Africa during the 19th century (Coetzee 1988, p. 30; Lester 2001, p. 16).

The second factor to mention, then, concerns the fact that most of the early Scots to become ministers in the DRC were either former missionaries of the LMS in South Africa ( 1990; Sass 1956, p. 20) or they were aspirant missionaries seeking their first call after having completed ministerial training in Scotland ( 1990; Theal 1898, pp. 116–17). The best-known name among the candidates directly recruited from Scotland is that of Andrew Murray. This name is famous because his son and namesake would become a bestselling evangelical writer and a noteworthy promoter of Christian mission. However, the senior Andrew, originating from Aberdeen in Scotland, had been keenly interested in mission himself. He had contemplated various possible destinations around the world for living out this vocation, eventually rejected a call to Newfoundland, and then accepted the opportunity to move to South Africa to serve in the Dutch Reformed Church, which became his lifelong adopted home and church (Neethling 1909, p. 7ff; Aschman 1972, p. 40). At his pastorate in the Karoo settlement of Graaff Reinet, Murray maintained and fostered a strong missionary identity. The parsonage would become a welcoming lodge for traveling missionaries belonging to, for example, the LMS and the Paris Evangelical Missionary Society. Figures such as Livingstone, Moffat, Casalis and Arbousset, to name just a few, stayed over on occasion and mingled with the Murray family (Neethling 1909, p. 15). Murray himself was known for having abolitionary views and he had a strong interest in the salvation and wellbeing of the indigenous population (Aschman 1972, p. 40; cf. Neethling 1909, p. 34). Yet, what is really of interest is that he, and other Scottish ministerial recruits, possibly also understood the Dutch population, which made up the majority of the church membership in the DRC, to be subjects in need of missionary intervention, or at least evangelical persuasion. At any rate, the Scots ministers disagreed with their congregants on a number of crucial matters, most notably regarding the issue of slavery and the question of obedience to the laws of the British colonial government. When a proportion of the Dutch population, who would later in the 19th century increasingly identify themselves as Afrikaners, decided to migrate north and eastwards, in order to escape the strictures of imperial legislation, this created much tension with their Scottish pastors and the DRC official structures (Dreyer 1929, p. 6). The church opposed the migration, but once it became clear that there was no chance of luring their recalcitrant flock back to the confines of the colony, the Scottish ministers again adopted a missionary approach and paid visits to these migrants and their eventual descendants through evangelistic tours crisscrossing the interior of what would eventually become Boer Republics (Dreyer 1899, p. 48; *De Kerkbode 1927*, March 30, p. 432), before these were finally incorporated into the Union of South Africa in 1910 as part of the British Empire.

### 3. Missionary Interests and the Germination of Apartheid

The well-known Andrew Murray Jr, second son of the abovementioned Andrew Murray from Aberdeen, after completing his ministerial training, was inducted at his first pastorate, located in the center of this northern territory. This was in the town of Bloemfontein (Du Plessis 1919, p. 77), which at the time was the northernmost parish of the DRC in what was then the Orange River Sovereignty, a territory loosely administered by the British for less than a decade, until it was relinquished by the Empire in 1854. This was much to the consternation of Andrew Murray who subsequently availed himself to be part of a two-man protest mission to the British parliament against the imperial withdrawal (Ibid, p. 152). Ultimately, this mission to England that Murray undertook in the company of one other delegate on behalf of the pro-imperial party in Bloemfontein was unsuccessful. The Orange River Sovereignty was abandoned and duly transformed into the Boer controlled Orange Free State (Ibid, pp. 155–56). Shortly thereafter, Murray resigned his pastoral position in Bloemfontein in order to take up a pastorate in Worcester, a town in the still British controlled Cape Colony.

However, during the years Murray spent in Bloemfontein, he used the opportunity to undertake several lengthy evangelistic tours to the Boer controlled Transvaal, an area to the north of the Orange River Sovereignty (Hopkins 1972, p. 730; Gerdener 1934, pp. 49–52). These tours, involving much preaching and sometimes other pastoral roles such as officiating at marriages and administering the sacraments, illustrated both his spiritual commitment to the Boer (Afrikaner) people as well as the presumed fact that Murray perhaps like his father and other Scots ministers in the DRC tended to view these Boers as their missionary subjects (See, Du Plessis 1919, p. 125). In any other global context, such a case of colonial-era white missionaries evangelizing other white subjects might have been anomalous, and indeed in South Africa, with its exponentially hardened racialized context, this would have been even stranger. However, I am not suggesting that such activities would have been openly acknowledged as part of a missionary enterprise. Mission, at least in as far as it was officially conducted, in most cases consisted of whites evangelizing blacks. Certainly, this was the norm in South Africa. So, the very fact that there might have existed a situation, perhaps best described as a subtext or a hidden discourse, of Scots missionary interested ministers evangelizing white self-avowed Christian Boers might have been perceived as a scandalous situation by the latter, had they been aware of the subtext. To some extent, it is probable that they were indeed cognizant of their implied status as missionary subjects. This would at least explain the occasional incendiary welcome which Boer communities offered to Andrew Murray during his evangelistic tours (Du Plessis 1919, pp. 122–23). It might even help to explain the growing race consciousness among these very same communities as the 19th and early 20th centuries progressed, to the point where apartheid became officially sanctioned as a policy by the DRC. White Afrikaner Christians for the most part wanted to make sure that there could be no question of equalization in the religious setting between them and their black brethren in the Spirit. Was such bigotry perhaps, at least in part, a backlash against their own implicit experience as missionary subjects to their foreign birthed and/or trained ministers, like the Murrays? It is a likely deduction that this was a contributing factor in the development of ecclesiastical apartheid.

I do not have the space here to offer a detailed exposition of the emergence of ecclesiastical apartheid in the DRC, but, given the case of Andrew Murray Sr., above, it might be worth pointing out that this patriarch of the Murray family in South Africa also has the dubious credit of being responsible for introducing a motion at the 1857 synod of the DRC (Sass 1956, pp. 138–39), which has been widely cited for being the first official procedure towards the institutionalization of church-based apartheid, which in turn would eventually serve as a kind of blueprint for the secular policy of apartheid (Loff 1983; Smit 2009, p. 461).

In fact, although the above has often served as a convenient shorthand explanation for the emergence of apartheid, the reality is rather more complex. Andrew Murray and his missionary enthused compatriots had been all too aware that there existed strong racial prejudice among their Dutch Afrikaner flock. Hence, when freed slaves and indige-

nous Africans converted as a result of missionary activities, there developed an increased resistance among the white church membership who tended to view the DRC as 'their' church, i.e., as a Dutch cultural bastion in the African wilderness. Some objected to sharing communion with 'heathen' converts, and in some congregations the very principle of communal worship, which had been accepted as the Christian standard, increasingly came under attack. So, Andrew Murray Sr., apparently fearing a schism in the church after a synodical debate on this contentious issue, tabled the abovementioned motion, accepted by a large majority, which stated unambiguously that although it was scripturally sound and otherwise orthodox for all Christians in a community to be congregated together for worship, if the 'weakness of some' made such an arrangement impossible, then separate church services could be allowed (Sass 1956, pp. 141–42).

With this accommodationist solution, Murray and his missionary minded colleagues must have felt satisfied that they managed to both ensure the continuance of missionary enterprises targeting the black population as well as protecting the DRC from undergoing a schism. Ironically, the measure in fact ended up normalizing racial segregation in the religious setting, with one consequence being that a couple of decades later the Dutch Reformed Mission Church (DRMC) would be founded in 1881, which was a separate ecclesiastical entity for black congregants, but under white supervision and financial control (Loff 1983, p. 22). In this way, the DRC became for all practical purposes a racially segregated church. Furthermore, this paternalistic structure of a DRC 'mother' and a DRMC 'daughter' provided a handy foundation and apparently a Christian sanction for the development of secular political apartheid in the next century (Elphick 2012, p. 222ff).

## 4. Scottish Afrikaners and Scots in Central Africa

With that, the scene has been set to address the question of the DRC's foreign missionary enterprise, which started in central Africa, Nyasaland (Malawi) to be exact. By the final decades of the 19th century, Andrew Murray Jr. and his pro-missionary peers were able to establish a Ministers Missionary Union that endeavored to fund and support a fledgling missionary enterprise emerging out of their own ranks (Du Plessis 1924, p. 15). As if in illustration of the close-knit nature of this MMU within the context the wider DRC, one could mention the name of the first candidate to avail himself to serve as missionary within the auspices of the MMU. That was Andrew Charles Murray, who was a nephew of Andrew Murray Jr and of course a grandson of the abovementioned Andrew Murray Sr. As a result of negotiations initially started between Andrew Murray Jr and James Stewart of the Free Church of Scotland's Livingstonia Mission in Nyasaland, it was agreed that AC Murray, who arrived there in 1888, would take responsibility for a mission field in the central area of that country (Murray 1897, p. 6ff). Initially, this mission headed up by AC Murray and funded by the MMU continued to fall under the supervision of Livingstonia, until the DRC finally took full responsibility for it more than a decade later. In 1903 it came under the supervision of the DRC's mission commission (Murray 1931, p. 120).

The early 20th century was an auspicious time for South Africa. After their loss to Britain in the South African War (1899–1902), the two Boer republics, the Orange Free State and the Transvaal, became coopted into the Union of South Africa as part of the British Empire. Yet, rather than extinguishing the fire of Afrikaner nationalism, the lost Boer cause fanned its flames to blaze in even wider directions (Boje and Pretorius 2011, pp. 59–72). Cape Afrikaners increasingly beat their drums to the nationalist tune, and the DRC, formerly controlled to a large extent by the Murrays and their Scots compatriots, became much more thoroughly aligned with the emerging Afrikaner nationalist civil religion (Moodie 1975). One consequence of this developing sense of ethno-national identity was the need to compare and compete with what was perceived as culturally respectable social formations. Hence, the DRC's foreign mission enterprise, formerly virtually a Murray family affair, now became a project of wider interest. During the course of the South African War and in its aftermath impressive numbers of missionary candidates were recruited and many of these received appointments in central Nyasaland (Kok 1971), where the DRC's Nkhoma mission increasingly asserted

itself in distinction from its Scottish partners to the north and south, where the Free Church of Scotland's Livingstonia mission and the Church of Scotland's Blantyre mission respectively operated. Yet, despite this more general influx of DRC missionaries, it is noteworthy that the Murrays and their sphere of influence retained the leadership positions in the mission field (Parsons 1998, p. 26). This occurred even as the DRC back in South Africa steadily moved away from its Scottish-influenced heritage in order to embrace more distinctively Afrikaner nationalist protagonists and ideals (Ross 1987, pp. 201–22).

In Nyasaland, the DRC's Nkhoma mission prided itself on its expanded activities, which included involvement in agriculture, industry, and especially medicine and translation work. Bible translation was one area where the DRC's William Hoppe Murray, cousin of the abovementioned AC Murray, played a leading role in collaboration with representatives of the Scottish Blantyre mission to the south (Murray 1931, p. 28; Livingstone 1931, p. 160).

So, to briefly pause before plunging ahead with the narrative, it would by now be clear that mission interest among one sector within the DRC was a factor that variously influenced broader opinions regarding race and separatism within the wider Afrikaner Reformed community. Mission grew both in popularity and controversy, and it would strongly influence theological and ultimately political discourse among the wider DRC in South Africa. Hence, mission was even in this context set to become a mother of sorts to theological developments, to paraphrase the abovementioned Martin Kähler. This would have a direct effect on church and society. Thus, far the central argument of the special edition regarding an ongoing conversation among theological history, mission history, and church history is thus sustained.

## 5. A Rocky Road towards a Compromised Ecumenism

With respect to the various activities of the DRC's Nkhoma mission, there existed much cooperation between the two Scots missions and the DRC mission, which was of course itself an offspring of the Scottish Livingstonia mission. Yet, the most important and ultimately most controversial collaborative project was the formation of a countrywide indigenous Reformed/Presbyterian church. Talks concerning the formation of such a church started early in the century between the two Scottish missions and over time the DRC mission became drawn into the discussion. Out of a couple of decades-long deliberation the Church of Central Africa (Presbyterian) (CCAP) was finally born, with the DRC's Nkhoma mission having its churches join those of the two Scottish missions some years after the initial establishment of the CCAP (Pauw 1980).

The final amalgamation of the Nkhoma mission with the CCAP was a bumpy one despite the general good will and commitment of missionaries on both sides to successfully accomplish this feat of ecumenism. Some tense moments occurred at a 1924 conference at Livingstonia, which was convened for the purpose of nailing down the exact terms of unity. The DRC's delegation consisted of Nkhoma missionaries and local church leaders as well as representatives of their church in South Africa. According to a couple of different sources, proceedings went well until mealtime arrived, when at least one unnamed DRC member discovered to his shock and horror that seating had been so arranged that black and white representatives would eat together, alongside shared tables (Retief 1948, p. 234; Murray 1924a). Such an arrangement allegedly contravened a racial tendency common among many Afrikaners of imposing segregation in social settings particularly where intimate rituals such as meals were concerned. The DRC contingent, or at least some of their prominent members, now balked at the arrangement and although they sat down and ate at their assigned seating spots, the meal was followed by some heated discussions between Scots and Afrikaners with opposing views regarding the correct social decorum surrounding meals for Christians from diverse racial backgrounds. In one reported discussion between W.H. Murray and the Livingstonia missionary, Donald Fraser, accusations of hypocrisy were flung back and forth, because although the latter found the former's attitude regarding racially mixed eating hypocritical, it seems upon further debate it allegedly

emerged that Fraser too had limits to the amount of equalization he would tolerate within his own household. This was specifically when it came to the hypothetical marriage of his daughter to an African, which he allegedly could not countenance. Such was the tone of the discussion according to W.H. Murray's biographer, at least (Retief 1948, p. 234). A.C. Murray, the founder of the DRC's missionary enterprise in Nyasaland/Malawi, in turn described the mixed eating incident and its aftermath in his diary, and there he mentioned a discussion that he had with Livingstonia missionary, McAlpine, who described the DRC's perspective on the controversy as 'unchristian' (Murray 1924b).

This has not become a widely published incident, apparently for the reason that A.C. Murray had urged his fellow representatives to not make much of this event when writing home (Murray 1924c). Murray was evidently worried that the tide of public opinion within the grassroots DRC back in South Africa would turn against church union in Nyasaland once people back there became aware of the racially relaxed norms holding sway among their Scottish missionary partners.

As for the progression of the 1924 conference, it seems the different contingents were able to set their differences aside for the purposes of the mutual goal of church unity, which did occur as a result of decisions at this conference. In fact, if DRC sources are to be believed, their own view prevailed in the sense that the other participants eventually agreed to accommodate the DRC's insistence on segregated eating arrangements for the remainder of the conference. It seems that the DRC's negotiating partners were of the opinion that the DRC's racial prejudice was a lesser evil than the potential derailment of the process of their Nkhoma's synod's amalgamation to the CCAP. Similarly, perhaps following A.C. Murray's plea in favor of cautious reporting, the Nkhoma mission representatives were able to convince their home DRC synod in South Africa to approve their process of joining the CCAP (Retief 1951, pp. 198–206), despite some lingering reservations regarding Scottish 'liberalism' from some in the DRC.

## 6. The Afrikaner Fear of Liberalism

This fear of Scottish 'liberalism' functioned on two levels. Theologically, the Scots, their missions, and ultimately the CCAP, were suspected of being susceptible to theological modernism. Significantly, the 1924 Livingstonia conference and its aftermath occurred in the context of a rising controversy among the DRC in South Africa that played out as a local variant of the modernist-fundamentalist conflict which erupted in North American protestant churches and on seminary campuses in the 1920s. A number of upcoming DRC theologians had studied at 'fundamentalist' schools in the USA, and from within this group a strong impetus emerged seeking to purge their local DRC in South Africa from its own 'modernist' influences. The target in chief of this pressure group was the Stellenbosch biblical scholar and missiologist, Prof. Johannes du Plessis (Erasmus 2009, p. 332). The Edinburgh educated du Plessis accepted and mildly propagated some of the results of higher criticism in biblical studies, insisting for example that, contrary to the prevailing belief among his DRC peers, that the Pentateuch had not been written by Moses, and that the book of Jonah could not be taken as historically factual (see Kerksaak 1931). Du Plessis, however, was roundly denounced in the South African theological community, primarily by American trained theologians who had studied at 'fundamentalist' seminaries, and also by Dutch trained disciples of Abraham Kuyper. Representatives from both of these groups insisted on a literalist interpretation of scripture and objected to Du Plessis's propagation of higher criticism (Coetzee 2010, p. 170ff).

After a protracted battle in church setting and ultimately in the civil courts, du Plessis had no option but to relinquish his position at the Stellenbosch Seminary. At least, after the Cape High court cleared him from wrongdoing the church had no option but to keep paying for his salary and pension, but he remained barred from returning to his teaching position, meaning that the so-called fundamentalists, or 'propositionalists', as Mieke Holkeboer variously refers to them (Holkeboer 1995, p. 24ff), won the ideological battle even if they had lost the civil case (Mouton 2009, p. 441ff). What makes du Plessis significant in the context

of this essay is the fact that he was the abovementioned Andrew Murray Jr's protégé, author of the latter's biography (Du Plessis 1919), and a leading missiologist in his own right (see Du Plessis 1965). As Holkeboer pointed out in a fine Master's thesis on a related topic, du Plessis never considered himself a theological liberal, but saw himself instead as fully in line with Murray and others who had fought their own court and ecclesiastical battles against a groups of 19th-century Dutch trained 'liberals' in the Cape DRC (Holkeboer 1995, p. 69). Obviously, liberalism, just as fundamentalism or evangelicalism, is a slippery term that meant different things in different contexts. What could be said definitively is that in the context of the DRC of the early 20th century this was not a mantle that any theologian wanted to wear voluntarily. Du Plessis' opponents attempted to paint him as such, and they were remarkably successful among a large proportion of the Afrikaner Reformed population. This was despite the fact that he was a genuine evangelical and missionary leader of note. Du Plessis was a close friend of DRC missionary WH Murray and he was an avid supporter of the Nkhoma mission (Erasmus 2009, pp. 335, 389). This meant that he was well aligned with the Nkhoma mission's collaboration with the Scottish missions and their eventual amalgamation with the CCAP. However, with his ultimate failure to remain in his teaching position, the fundamentalists and Kuyperian neo-Calvinists were in the ascendancy within the theological landscape in the DRC.

When du Plessis lost the struggle, the Nkhoma mission and their participation in the CCAP were in a much more vulnerable position. One of du Plessis' most ardent opponents was a former missionary to West Africa and subsequently the mission secretary of the DRC in the Orange Free State, J.G Strydom (Elphick 2012, p. 226ff). One of the strongest bones of contention for Strydom and those of his ilk was the fact that the CCAP had in its period of formation opted to forego any attempt at extensive doctrinal conformation to typical Reformed creeds, opting instead to keep things simple with a basic statement of Faith that everyone could easily subscribe to (Pauw 1980, p. 274). For Strydom, this statement was far too vague in its proclamation of the extent to which the Bible could be said to be the Word of God. It did not, for example, insist on the doctrine of scriptural inerrancy. This lapse, according to Strydom and his supporters in the DRC in South Africa, was indicative of the liberalism of the Scottish founders of the CCAP, a liberalism to which the Nkhoma mission had now all too readily acceded (Ibid., p. 280).

Furthermore, in the aftermath of the Nkhoma synod's joining up with the CCAP, another DRC mission, this time the *Madzi Moyo* mission in Northern Rhodesia (Zambia), considered the possibility of also amalgamating to this unified church (Ibid.). This was however, unlike the Cape DRC controlled mission of Nkhoma, a mission under the control and sponsorship of the Free State synod of the DRC, where Strydom was the mission secretary. This was probably the start of Strydom's interest in the CCAP, because he subsequently became part of a 1928 delegation by the Free State DRC's synod that toured Nyasaland and the DRC missions and CCAP churches there, in order to inspect the situation regarding (un)orthodoxies of varying kind. According to authoritative sources he was deeply shocked at the level of social mixing between Africans and white missionaries he encountered in that country, and this played a major role in his subsequent opposition to DRC participation in the CCAP (Ibid.). Strydom, it should also be pointed out at this point, was an Afrikaner nationalist and a noted early proponent of apartheid as a policy not only in church but also in South African society at large. He, for example, wrote at an early stage to DF Malan, the nationalist politician who would initiate apartheid as government policy when he became prime minister in 1948, to adopt a stricter policy of total separation than even Malan could countenance at the time (Badenhorst 1981, p. 150). It is a certain fact that much of the proverbial blueprint for apartheid as a policy of state was originally adapted from hardline DRC missionary policies, and Strydom himself played no small part in that as the above interaction with Malan would indicate. Increasingly, in South Africa as the 20th century unfolded, Strydom and his perspective of total racial separation became more and more normative in Afrikaner churches and in society until it all became formalized with the official institutionalization of apartheid in 1948.

### 7. When Racial and Doctrinal Heterodoxy Align

The tension between the nationalist, separatist views of Strydom, on the one hand, and the ecumenicity of the Nkhoma mission, on the other, became highlighted in a series of articles published in 1940 in the official DRC paper, *Die Kerkbode*, featuring something of a debate regarding the orthodoxy/heterodoxy of the CCAP (*Die Kerkbode* 1940a, 1940b, 1940c). This was really a back and forth between Nkhoma missionary, J.A. Retief, on the one side, and Strydom, on the other. It was set off by what for all intents and purposes seemed like a fairly innocuous article by the Nkhoma missionary in which Retief explained to the general readership of *Die Kerkbode* in South Africa the nature of the relationship between church and mission in Nyasaland and especially emphasizing the benefits of church union in the form of the CCAP. To this, Strydom immediately penned a response wherein he lodged a number of complaints against the CCAP and in the process contradicted every argument advanced by his colleague, Retief, in defense of church union. To Strydom, there were clear and irreconcilable differences between the DRC and the Scottish churches involved in the CCAP, so much so that the very foundations of the church unity project were extremely shaky. While the DRC in Strydom's view was Reformed and orthodox, the Scottish churches and missions were characterized by liberalism which according to Strydom's perspective was apparently just another word for heterodoxy. Moreover, unlike the Scots, who had the opposite policy, the DRC, according to Strydom's view was characterized by "*total social apartheid* between whites and colored races of Africa" [transl.; his italics] (*Die Kerkbode* 1940b).

### 8. Conclusions

It is perhaps ironic that Strydom's view, which would steadily become the mainstream view in South Africa's DRC, would posit the twin orthodoxies of Reformed Calvinism and 'total social apartheid' as equally non-negotiable. In hindsight it is a combination of ideologies that would make many Reformed Calvinists in South Africa and indeed elsewhere cringe to say the least. It was a view that explicitly placed ethnic nationalism on a par with religious orthodoxy, which in fact refused to countenance any separation between the two themes. Strydom was just one notable proponent of this perspective in Afrikaner history. Nonetheless, with these types of notions he became one of the early instigators of an alternative theology of apartheid, which in turn became a predominant hermeneutical lens through which a generation of Afrikaner nationalist theologians and ministers would interpret scripture in South Africa. It was a strange development to be sure, but perhaps also not so unexpected if we consider, for example, the 'alternative facts' underpinning much of the political discourse influencing contemporary evangelicalism, most strikingly in the USA, where a large segment of the evangelical movement has become joined at the proverbial hip with white nationalism (Fea 2018).

The Nkhoma mission, which had become part of the Nkhoma synod of the CCAP, continued to traverse its own tightrope between the ecumenism their members had opted for and the exclusivist demands of the missionaries' home church, which was also the mission's main funding source. While a leading Nkhoma missionary such as the above-mentioned J.A. Retief would vociferously defend the CCAP against the aspersions of J.G. Strydom on the pages of *Die Kerkbode*, it is interesting and revealing that Strydom's basic premise regarding the orthodoxy of 'total social apartheid' would not be disputed by his Nkhoma opponent in this debate. To the contrary, Retief's defense of the CCAP and their Scottish missionary partners proceeded from an argument that Strydom overstated the liberality of social mixing in the CCAP and that the Nkhoma mission was in fact in the process of turning the other missions around to views more reflective of the DRC's position. This was a disingenuous argument, which missionary Retief no doubt hoped would be believable to *Die Kerkbode*'s readership, a readership which broadly shared the views espoused by Strydom and other apartheid apologists, but a readership that would have very little knowledge of what actually went on in the Nyasaland mission field beyond

what was reported to them through publications such as *Die Kerkbode* and other similar outlets.

The upshot was that Nkhoma managed to maintain its affiliation to the CCAP despite the fact that the DRC in South African plunged ever deeper into apartheid doctrine as the 20th century proceeded. However, it might be fair to say that whereas this mission had formerly been a source of pride to the home church, the Nkhoma missionaries found themselves increasingly marginalized within the prevailing discourse in the DRC of the mid-20th century and beyond. The Nkhoma synod within the CCAP did not remain free from the politics of its sponsoring body. Although 'total social apartheid' was never a realistically enforceable option in a church in a country such as Malawi that was almost entirely comprised of black Africans, the conservative theological views that otherwise prevailed in the DRC of South Africa surely had more of a long-term impact. One example to conclude with is the fact that whereas both Livingstonia and Blantyre synods had over time come to accept the ordination of women in the preaching ministry, the Nkhoma synod continues to resist any such regulation to this day (Hofmeyr and Munyenyembe 2017, pp. 9–12). Women remain barred from ordination to word and sacrament, an historical legacy which is extremely ironic in the light of the fact that the DRC in South Africa had officially changed its own policy regarding this matter, and of course also regarding its former adherence to apartheid, several decades ago already.

This case of the DRC and the CCAP illustrate well how mission, theology, and church history have hung together in constant conversation in these various contexts. While missiologists and proponents of Christian mission might wish for missionary activity to be shown as a positive influence in human history, this particular case study presents a messy picture, which bears out my thesis. On the one hand, this narrative shows how mission when tied to ethnic nationalism might lead to a theology of apartheid. On the other hand, it shows that when mission prioritizes ecumenism it opens the possibility for ideological cracks to appear within a nationalistic armor. Ultimately, the case of the DRC's participation in the CCAP presented the potential for ideological cracks to form but for those to have widened into anything that could significantly threaten the construction of Afrikaner Christian nationalism in South Africa, the cross-infusion of ideas from the CCAP in Malawi to the DRC in South Africa would have had to be stronger than it ultimately was. In the end, the larger, wealthier church, the DRC, in the more powerful country, South Africa, were able to shrug off any such challenge from its periphery. For most of this intertwined history, influence seemed to move more decisively in only one direction, from South Africa to Malawi, rather than the other way around. As such this presents itself perhaps as a kind of microcosm of North–South relations on a global scale, despite apartheid South Africa, being geographically to the south of Malawi, in this case representing the ideological North. Yet, we know that potentiality could easily be turned into actuality, and hence the threat of a periphery to the ideological walls of a center should not be underestimated even if, as in this individual historical case, there are no obvious, overt successes to show for how peripheral developments might have overturned perceptions back in the center.

Within the broader frame of the special edition, this case study illustrated initially how mission interest and eventually missionary activities influenced the trajectory of the DRC's history in South Africa in various ways. Then it showed how an emergent theological history within the DRC, a theological history based on narrow confessionalism and Christian nationalism ended up influencing the church's missionary movement to the point where it nearly derailed an ecumenical venture involving the Scottish missions in Malawi. Finally, it was pointed out how this theological history emerging from South Africa continues to influence church policies in the CCAP of Malawi in insidious ways.

Let me conclude by stating that case studies such as this one might offer a lens through which the constant conversation between church history, mission history, and theological history might be studied. Furthermore, if one takes this ongoing conversation as a premise, as I have done here, then it might illuminate a well-trodden history, in this case the history of apartheid and the Dutch Reformed Church's involvement in it, in new

and even unexpected ways. Mission, in this case, is revealed not so much as a mother to theology, but as being in constant symbiotic relationship with theology, creating thereby a powerful synthesis that could decisively shape church and society.

**Funding:** The research received no external funding.

**Institutional Review Board Statement:** Not applicable.

**Informed Consent Statement:** Not applicable.

**Conflicts of Interest:** The author declares no conflict of interest.

## Abbreviations

| | |
|---|---|
| CCAP | Church of Central Africa (Presbyterian) |
| DRC | Dutch Reformed Church |
| DRMC | Dutch Reformed Mission Church |
| LMS | London Missionary Society |
| MMU | Ministers Missionary Union |

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
