# Peer review of "Traversing a Tightrope between Ecumenism and Exclusivism: The Intertwined History of South Africa’s Dutch Reformed Church and the Church of Central Africa Presbyterian in Nyasaland (Malawi)"

_religions, doi:10.3390/rel12030176_

Round 1

Reviewer 1 Report

This is a very interesting paper. It presents a topic of great relevance both in its geographical relations (South Africa and Malawi) and its theoretical connections (theology, missiology, and history of Christianity). Indeed, I think missiology (rather than history of missions) is probably a better identifier, as the former concerns matters of context and culture, as well as philosophy of transmission. The theoretical framework for linking these disparate discourses is not clear, at least as far as I can see. 

While he is not mentioned, theologian John de Gruchy has attempted this kind of interdisciplinary dialogue using social history and the work of the Comaroffs on colonial discourse as well as more familiar church historical sources. See Christianity and the Modernization of South Africa (Unisa, 2009). De Gruchy also has coverage of the Du Plessis trial on pp. 135-137 that presents a more nuanced picture than the present article. Another work that might be suggestive, especially on the doctrinal-theological side is an excellent thesis by Mieke Holkeboer: "Before the Pool of Narcissus: The Nederduitse Gereformeerde Kerk's Journey to Confessional Orthodoxy and Isolation Through the Lens of Doctrine." (UCT, 1995). She uses the "cultural-linguistic" theory of doctrine of George Lindbeck. (The term "cultural-linguistic" is used in the present article on p. 2, but in a different context). 

One of the things de Gruchy and Holkeboer provide nuance to is the category of liberalism, a notoriously slippery term when applied across doctrinal, ecclesial, and historical-geographical discourses. As Irving Hexham showed in his study of the Gereformeerde Kerke the term captured both English political and social progressivism and anti-Calvinism (or "Methodism"). It certainly meant different things in North America and South Africa, as I think the work of Richard Elphick shows (referenced in the article but not used in this regard). Even within South Africa it could (and can) be used as description or accusation. 

Another term I was unsure about is "fundamentalism" (pp. 7-8). Used in the early twentieth century it has specific reference to "The Fundamentals: A Testimony to the Truth" published in the United States between 1910 and 1915. Used today it is a synonym for Right Wing Christianity... or really anything considered "too conservative." Hence the unhelpfulness of the term. A term like "Reformed Scholasticism" (de Gruchy) is one possible alternative, though it also may be unsatisfactory. Dutch neo-Calvinism in South Africa was certainly conservative. But fundamentalist seems to me the wrong term.

Du Plessis is a fascinating person, and constitutes one  translation of the Murray legacy into the early twentieth century. What is particularly interesting is the way that under his leadership DRC missions was deeply ecumenical, and took initiative within South Africa at least in fostering inter-church and inter-agency collaboration. Bearing in mind the legacy of Cape liberalism (that word again) was about the "uplifting" of the races, it is arguable that the vision of separate churches (c.f. Henry Venn's "Three Self" missiological programme) could be viewed as progressive or regressive (as in "total social apartheid") depending on where one stood. He was a victim of the post SA War "ethnic mobilization" of Afrikaners and the idea of the volkskerk (a term I don't see in the article) which seeped into the DRC. 

In this regard, the phrase "hardline DRC missionary policies" probably should also be softened or at least nuanced. There are other terms that should be qualified as well, for instance the "Scottish infused heritage" of the DRC (influenced... yes; but infused...?). The use of the term "Afrikaner" before the twentieth century may also need qualification. Even applied to 1924, the identification of "the Afrikaner racial custom of imposing segregation in social settings..." is an overly general statement. And did "the fundamentalists and Kuyperian new-Calvinists [take] full control of the theological landscape in the DRC" in the wake after Du Plessis? Was the DRC in the 1930s really that unified?

For those reading without a knowledge of the distinctive history of South Africa it may be worth noting that "slavery" in the Cape was dominated by the importation of labour from the Indonesian archipelago, rather than the enslavement of indigenous Africans. Though perhaps outside the scope of the paper, it's interesting that the VOC prohibited the enslavement of Christians, which was certainly no encouragement to evangelism!

I would eagerly anticipate seeing a finished version of this article. It would make an important contribution to theology, missiology, and history if it were to discipline and nuance its language and provide a more clear theoretical base for the interdisciplinary points. It also needs some editing for long sentences (including in the Abstract) and general clarity of expression.

Author Response

Thank you very much for a very helpful review. It has among other things introduced me to the important thesis of Mieke Holkeboer, which I had known seen before. I found it so interesting that I read it in its entirety.

Regarding the critical points mentioned, the following:

  1. The theoretical framework might not have been clear because I simply took the special edition's theoretical position regarding the constant conversation between mission history, church history, and theology as the basis according to which I then attempted to illustrate the interactions between such elements within this particular case. The special edition's description does not mention missiology as such, which is why I avoided theorizing along those lines. I have however now added further grounding in this respect in reference to Martin Kahler.
  2. I have much appreciation for de Gruchy's perspective, but within a short and narrowly focused article such as this one, I believe the addition of social historical themes such as modernization would have drawn the attention unnecessarily away from the particular narrative I wanted to highlight, which is already a complex enough story, given that it involves two different countries and churches over the course of a century.
  3. I agree with the critique regarding the use of liberalism, fundamentalism, etc. I have added nuance and further qualification. However, I use the term fundamentalism exclusively in its historical sense, in other words in the context of the modernist-fundamentalist controversy in the United States in the 1920s, and in the way 'fundamentalist' ideas during this same period was exported to South Africa by theologians who had studied at then fundamentalist schools. These were the main challengers to du Plessis, along with Kuyperian neo-Calvinists. I agree that fundamentalists and neo-Calvinists were not the same, and that they should not be conflated, but as Murray Coetzee indicated in his thesis on this subject, the two groups combined forces in their opposition to du Plessis and his followers.    
  4. Lots could be written regarding du Plessis of course, and I have added some perspectives particularly in reference to the extensive treatment he receives in Holkeboer. However, despite how interesting this is, in this article du Plessis is less of a central character than he might typically be. The focus is on the intertwined contexts of Malawi and South Africa and especially the ecumenical project in Malawi and its repercussions in South Africa. As it is, du Plessis was not actively involved in the discussions surrounding the CCAP. So, his story, although important for context, is more of a back story in this case.
  5. The point about slavery is an important one, but as the reviewer points out, explaining the state of the institution and all of its implications in the VOC might have taken a different trajectory that what is attempted here. So, instead, I chose to simplify my reference to slavery, to at least contain the potential for confusion as far as possible. 

Reviewer 2 Report

This is a strong and original article, subject to some minor editing.  If possible, you should make some reference to the recently published Kenneth R. Ross and Klaus Fiedler, A Malawi Church History (Mzuni Press, 2020).  Some minor points:

line 87: correct spelling to John Philip

lines 164-5:  Mission did not 'always' consist of whites evangelising blacks.  What about the Xhosa missionaries to Malawi written about by Jack Thompson?  Or African American missions to West Africa?  And the European churches most active in foreign mission were also heavily involved in mission to those regarded as the 'domestic heathen' in their own countries.

line 453: 'scot free' is an unfortunate phrase in view of the subject of the article, or is this a deliberate pun?

lines 469-70: 'this narrative shows how mission when tied to ethnic nationalism almost naturally leads to a theology of apartheid'.  This generalisation needs some modification.  Certainly the marriage of mission and ethnic nationalism is liable to lead to all sorts of problems, but the theology of apartheid is only one example of these problems.  Whites are not the only ones to have allowed mission agendas to be compromised by ethnic nationalism. 

The style of the article would be much improved by the removal of all the redundant and distracting references to yourself as author - what you think, believe, have or have not done in the article: lines 156, 169, 178-9, 181-2, 216, 221, 260, 495, 498.

Author Response

Thank you very much for this helpful review. I have attended to these points of critique and revised the article accordingly.